# Moderating Effect of Muscular Strength in the Association between Cardiovascular Events and Depressive Symptoms in Middle-Aged and Older Adults—A Cross Sectional Study

**DOI:** 10.3390/geriatrics9020036

**Published:** 2024-03-12

**Authors:** Diogo Veiga, Miguel Peralta, Élvio R. Gouveia, Marcelo de Maio Nascimento, Laura Carvalho, Jorge Encantado, Adilson Marques

**Affiliations:** 1Centro Interdisciplinar de Performance Humana, (CIPER) Faculdade de Motricidade Humana, Universidade de Lisboa, 1499-002 Cruz-Quebrada, Portugal; dmcveiga@fmh.ulisboa.pt (D.V.); mperalta@fmh.ulisboa.pt (M.P.); laura.carvalho@e-fmh.ulisboa.pt (L.C.); jencantado@fmh.ulisboa.pt (J.E.); 2Instituto de Saúde Ambiental (ISAMB), Faculdade de Medicina, Universidade de Lisboa, 1649-026 Lisboa, Portugal; 3Department of Physical Education and Sport, University of Madeira, 9020-105 Funchal, Portugal; erubiog@staff.uma.pt; 4Laboratory for Robotics and Engineering Systems (LARSYS), Interactive Technologies Institute, 9020-105 Funchal, Portugal; 5Department of Physical Education, Federal University of Vale do São Francisco, Petrolina 56.304-205, Brazil; marcelo.nascimento@univasf.edu.br

**Keywords:** depression, elderly, grip strength, moderation, cardiovascular diseases

## Abstract

Background: Depression and cardiovascular diseases are two main health conditions contributing to the global disease burden. Several studies indicate a reciprocal association between them. It is still unclear how changes in overall muscle strength may impact this association. This study aimed to analyse how muscular strength moderates the relationship between cardiovascular events and depressive symptoms among middle-aged and older adults. Methods: Wave 8 of the population-based Survey of Health, Ageing, and Retirement in Europe (2019/2020) cross-sectional data, which included 41,666 participants (17,986 men) with a mean age of 70.65 (9.1) years old, was examined. Grip strength was measured twice on each hand using a dynamometer. The 12-item EURO-D scale was employed to gauge depressive symptoms. Results: Grip strength negatively moderates the link between cardiovascular events and depressive symptoms (male: B = −0.03, 95% CI = −0.04, −0.03; female: B = −0.06, 95% CI = −0.06, −0.05). Additionally, the grip strength moderation values in the significant zone for males and females were less than 63.2 kg and 48.3 kg, respectively. Conclusions: Muscular strength modifies depressive symptoms and lessens their correlation with cardiovascular diseases. Muscle-strengthening activities could be incorporated into primary and secondary preventive strategies to reduce the burden of depression in people with CVD.

## 1. Introduction

Depression and cardiovascular diseases (CVD) are two main health conditions contributing to the disease burden globally [1]. Depression can harm mental and physical health, hamper daily activities, and negatively impact the quality of life [2]. According to the Diagnostic and Statistical Manual of Mental Disorders, Fifth Edition (DSM-5), the diagnosis of depression requires five or more symptoms to be present within a 2-week period [3]. One of the symptoms should, at least, be either a depressed mood or anhedonia. The secondary symptoms are appetite or weight changes, sleep difficulties, psychomotor agitation or retardation, fatigue or loss of energy, diminished ability to think or concentrate, feelings of worthlessness or excessive guilt, and suicidality. Although depression is one of the top causes of disability worldwide, most people who have depression still do not obtain effective treatment [4]. Although there is significant variance between countries, the lifetime risk of depression is (at least) 10% [5,6], and in patients with CVD, depression is estimated to be one in five [7]. According to statistical reports, nearly 45% of patients with CVD struggle with major depressive disorder [8]. Furthermore, having comorbid depression is independently linked to a worse prognosis for people with CVD [9].

Both CVD and depression share common pathogenic mechanisms [10]. Increased levels of inflammatory markers [11], abnormalities in sympathetic and parasympathetic activity [12] and irregularities in hormone and neurotransmitter levels (such as cortisol and serotonin) [13] have all been linked to depression and CVD. Additionally, there are data indicating that the co-occurrence of these diseases may be influenced by similar genetic variables [14]. Therefore, it is plausible that CVD is an important risk factor for depression. A cardiovascular disease diagnosis or ongoing condition can result in substantial emotional anguish, worry, and depression. Furthermore, mood and mental health can be directly impacted by the physiological effects of cardiovascular disease on the brain and body as a whole [15]. Depression is associated with physiological changes, including increased inflammation, altered autonomic nervous system activity, and hormonal imbalances, which can contribute to the development and progression of CVD [16]. Prior research has demonstrated a significant rate of co-morbidity between depression and CVD, indicating a reciprocal association between the two conditions [17,18,19]. Delivering physical exercise or physical activity may not only improve depression severity, but also directly tackle the constitutive elements of cardiovascular risk [20]. Several investigations have suggested that muscle strength is a plausible mechanism for these associations, as it has a protective effect on the development of CVD [20,21,22,23]. Also, both dynapenia (muscle weakness) and depressive symptoms are common in the elderly population [24], although women seem to live longer and have more years free of dynapenia than men [25]. Dynapenia is often accompanied by increased inflammation and oxidative stress. Chronic inflammation and oxidative stress are common features of both depression and CVD [26]. Among middle-aged and older adults, grip strength (GS) may mediate the relationship between depression and the estimated 10-year risk of CVD [27]. GS is a measure of overall body strength, and it is also a maximum hand static force measure commonly used to capture muscular strength and monitor the health status of the general population, especially in older individuals [28].

As the world’s population ages and life expectancy rises, it is increasingly important to consider middle-aged and older adults’ health, including depression and CVD. There is substantial complexity in the associations between depression, GS, and CVD. Although some of these links are understood, it is still unclear how changes in GS, as a proxy for overall muscle strength, may impact the association between CVD and depression. The research question was established: among European middle-aged and older adults of both sexes, does GS have a moderating role in the relationship between depressive symptoms and CVDs? Towards the hypothesis that GS weakens the link between depressive symptoms and CVDs among European middle-aged and older adults of both sexes, this study aimed to analyse how GS moderates the relationship between two major cardiovascular events, myocardial infarction and stroke, and depressive symptoms among middle-aged and older adults by sex.

## 2. Materials and Methods

### 2.1. Participants and Procedures

Data from wave 8 (2019/2020) of the population-based Survey of Health, Ageing, and Retirement in Europe (SHARE) served as the foundation for this research. The SHARE methodology was previously detailed in [29]. It is a biennial survey that gathers data from several European nations and Israel. The target population consists of all people living in residential households who are 50 years of age or older, plus their (possibly younger) partners. Those who do not reside at the sampled address (e.g., because it was a seasonal or vacation residence), are physically or mentally unable to participate, died before the start of the field period, or cannot speak the specific language of the national questionnaire, were excluded. The Ethics Council of the Max Planck Society for the Advancement of Science and the University of Mannheim Ethics Committee accepted the SHARE protocol. Written informed consent was obtained from all participants involved in the study.

A total of 41,666 participants (17,986 men and 23,680 women), with a mean age of 70.65 (9.1) years old, from 29 different countries (Austria, Germany, Sweden, the Netherlands, Spain, Italy, France, Denmark, Greece, Switzerland, Belgium, Israel, the Czech Republic, Poland, Ireland, Luxembourg, Hungary, Portugal, Slovenia, Estonia, Croatia, Lithuania, Bulgaria, Cyprus, Finland, Latvia, Malta, Romania, and Slovakia) made up the final sample.

### 2.2. Measures

#### 2.2.1. Depressive Symptoms

Depressive symptoms, the outcome measure, were assessed with the 12-item EURO-D scale. Scores vary from 0 to 12, with higher scores indicating more severe symptoms of depression. A cut-off of ≥4 points indicates clinically significant depression [30,31]. The validation and explanation of the scale are covered elsewhere [31].

#### 2.2.2. Cardiovascular Events

The exposure measure was being previously diagnosed with a cardiovascular event (stroke and/or myocardial infraction). Participants were asked to report being previously diagnosed with a cardiovascular event (stroke and/or myocardial infraction) by a medical doctor.

#### 2.2.3. Grip Strength

The moderator employed was GS. Using a dynamometer, it was measured twice on each hand (Smedley, S Dynamometer, TTM, Tokyo, Japan, 100 kg), switching between the left and right hand [32]. Participants held their upper arm tightly against their bodies while standing or sitting, with the elbow at a 90-degree angle, the wrist in neutral, and the inner lever of the dynamometer set to the hand. After practicing, participants exerted the dynamometer’s maximum pressure for 5 s. The GS variable contained the maximum value of the GS measurement of both hands. The values of two measures that differed by more than 20 kg were considered invalid. Measurements of GS that were equal to or more than 100 kg were excluded, as were measurements where GS was only assessed once in one hand.

#### 2.2.4. Co-Variables

Covariates included sex, physical activity, hypertension, age, and country, which were self-reported. Physical activity was measured as “frequency of moderate physical activity” (e.g., gardening, cleaning the car, going for a walk) and “frequency of vigorous physical activity” (e.g., sports, heavy housework, a job involving physical labour). The response alternatives for both moderate and vigorous activity were: (1) more than once a week, (2) once a week, (3) up to three times a month, and (4) hardly ever or never. The last two response options were grouped into one category called less than once a week.

### 2.3. Statistical Analysis

Descriptive statistics, including mean and standard deviation for continuous variables and frequency for categorical variables, were calculated. The *t*-test (for continuous variables) and chi-square (for categorical variables) were used to compare participants’ characteristics between sexes. An independent sample *t*-test and a Pearson correlation analysis were used to compare the depressive symptoms of men and women according to cardiovascular event diagnosis and to determine the relationship between GS and depressive symptoms. Based on the moderation methods suggested by Baron and Kenny [33], a moderated analysis of GS (moderator, W) on the connection between cardiovascular events (categorical, X) and depressive symptoms (continuous, Y) was conducted, and unstandardized coefficients were presented. The moderation analysis was carried out using Andrew Hayes’ PROCESS macro-3.5. The Johnson–Neyman method was used to evaluate statistically significant interactions and find regions of significance. This process was also used to determine a threshold of statistical significance. The analysis was stratified by sex and adjusted for age. Data analysis was performed using IBM SPSS Statistics version 28 (SPSS Inc., an IBM Company, Chicago, IL, USA) for Apple Mac^®^. For all tests, the statistical significance was set at *p* < 0.05.

## 3. Results

Table 1 presents the descriptive analysis. More women (30.5%) than men (18.3%) reported having depressive symptoms above cutoff. In contrast, more men (18.4%) than women (12.4%) reported having a history of cardiovascular events (myocardial infarction and/or stroke). A total of 6262 men and women had at reported least one cardiovascular event (5153 myocardial infractions and 1548 strokes in total).

The correlation analysis between GS and depressive symptoms showed that GS was significantly and negatively correlated with depressive symptoms for the total sample (r = −0.254, *p* < 0.001), as well as for males (r = −0.193, *p* < 0.001) and females (r = −0.210, *p* < 0.001) separately.

Table 2 compares depressive symptoms between participants according to the history of cardiovascular events (myocardial infarction and/or stroke). Regardless of sex, participants with a cardiovascular event had higher mean depressive symptoms than participants without CVD (males: 2.53 vs. 1.76, *p* < 0.001; females: 3.55 vs. 2.50, *p* < 0.001).

Table 3 presents the small moderating effect of GS (W) on the association between cardiovascular events (X) and depressive symptoms (Y). The link between cardiovascular events and depressive symptoms was negatively moderated by GS (male: B = −0.03, 95% CI = −0.04, −0.03; female: B = −0.06, 95% CI = −0.06, −0.05), meaning that greater GS led to a weaker association. The Johnson–Neyman test also revealed that the GS moderation values in the significant zone for males and females were less than 63.2 kg and 48.3 kg, respectively.

## 4. Discussion

This study aimed to analyse how GS moderates the relationship between two major cardiovascular events, myocardial infarction and stroke, and depressive symptoms among middle-aged and older adults. Results revealed that GS was positively related to lower depressive symptoms in middle-aged and older individuals. Furthermore, GS had a small moderating role in the association between CVEs and depressive symptoms in both men and women, possibly weakening its link to CVD.

Gender differences in depression have been extensively addressed over the past few decades. Our results support the evidence that women suffer more from depression than men (30.5% and 18.3%, respectively), which is in line with the majority of research that has found that women tend to experience depression twice as frequently as males do, independently of the culture [5]. The reasons for the disparities in depression between men and women include biological, psychological, and social factors [34].

Although using different methodologies, the negative correlation between GS and depression in this study matches other large-scale multinational studies carried out with older people [35,36]. The association of GS with depressive symptoms seems significant and inverse in individuals with and without chronic diseases, namely, CVD [37]. The findings also suggested that participants with a cardiovascular event diagnosis tend to present higher scores of depressive symptoms than those without it, both males and females. Depression seems to be highly prevalent in cardiac patients [38]. The fact that the longer a person lives with CVD, the greater the cumulative psychological impact [39], the functional impairment caused by CVD can decrease the ability to engage in desired activities and lead to depression [40], and intensive treatments, medications, or invasive procedures used to manage severe CVD can have physical and psychological side effects [41] could possibly explain how the duration and intensity of CVD can affect depression and vice versa. There may be a physiological connection between CVD and depression, as prior research demonstrates the close relationship between these two clinical diseases [42]. Although these relationships can be complex and multifaceted, several physiological mechanisms can also help explain the connections between these conditions and GS. Cardiovascular health is linked to GS through the muscle–heart axis (myokines production) [43]; high GS may indicate a lower overall inflammatory and oxidative burden (contributing to better cardiovascular and mental health) [44]; muscle strength exercises promote neuroplasticity and brain health, helping to mitigate the risk of depression by enhancing brain function and structure, as well as reducing the risk of cardiovascular diseases by improving cerebral blood flow and reducing vascular risk factors [45]. Still, most of the underlying mechanisms underpinning the connection between these two illnesses are unknown [46]. For instance, compared to people without CVD, CVD patients are more likely to experience depression.

GS moderated the relationship between depressive symptoms and cardiovascular events (myocardial infarction and stroke), slightly buffering this association for both males and females. According to the findings, people with cardiovascular events tend to suffer more from depression (presenting higher score values of depressive symptoms), and although the effect was small, having greater GS protected against depressive symptoms. It is important to note that, although the effect is significant, as it is small, intervention studies are needed that can analyse this moderation effect in terms of clinical practice. Nevertheless, once nearly 45% of patients with CVD present depressive symptoms [8] and it is estimated that depression is one in five in this population [7], improving muscular strength may be an effective non-pharmacological option to reduce the burden of depression in people with CVD.

According to the Johnson–Neyman test, this small moderation effect is present for GS values under 63.2 kg for males and 48.3 kg for females. In the elderly population, the GS measurement is more commonly used as a screen for sarcopenia, a very prevalent condition in this type of population. These results indicate that even for GS values lower than the cut-off for sarcopenia (<27 kg for men and <16 kg for women) [47], strength seems to be a moderator of depressive symptoms. Even for an elderly person who continues to be at risk of being diagnosed with sarcopenia, interventions that increase their strength may have a small protective effect against depression, attenuating its symptoms.

When analysing the current results, it is important to consider their advantages and disadvantages. As far as we know, this is the first study to examine the moderating role of GS in the relationship between depressive symptoms and the two major cardiovascular events, myocardial infarction and stroke. Additionally, this study used an objective measure of physical fitness (i.e., GS) as a moderator, assessed by trained personnel. Self-reported questionnaires are frequently used in large sample studies to quantify physical activity or even, in some cases, physical fitness, which typically yields higher values than objective measures [48]. Despite these strengths, several limitations must be acknowledged. First, the diagnosis of cardiovascular events, myocardial infarction and stroke, were self-reported, which may lead to memory bias. Also, we did not measure the number of previous CVEs and did not include the current medication of the participants. Second, the study’s cross-sectional design restricts the interpretation of the current findings in a cause-and-effect way, despite the large multinational sample. Lastly, another limitation was the inability to account for several confounding variables, including important risk factors such as the history of depression, coronary heart disease, cigarette smoking, and excessive alcohol consumption. Future research should consider testing more complex moderation models, where other variables are analysed that may also be playing a moderating effect on these relationships. Other covariates that were not included in this study should also be included in these models. In any case, these results reinforce the evidence that, in the clinical context, strength training could be integrated into rehabilitation programmes for this type of population.

## 5. Conclusions

GS, a proxy of muscular fitness, slightly buffers the association between cardiovascular events (myocardial infarction and stroke) and depressive symptoms. This supports the idea that physical activity, namely muscle-strengthening activities, may be incorporated into primary and secondary preventive strategies to reduce the burden of depression in people with CVD.

## Figures and Tables

**Table 1 geriatrics-09-00036-t001:** Characteristics of the total sample and divided by sex.

	Mean (95% CI) or % (95% CI)			
	Total (*n* = 41,666)	Male (*n* = 17,986)	Female (*n* = 23,680)	*p*-Value
Age (years)	70.65 (70.56, 70.73)	71.12 (70.99, 71.25)	70.29 (70.17, 70.41)	<0.000
GS (kg)	32.04 (31.93, 32.14)	40.68 (40.54, 40.83)	25.47 (25.38, 25.55)	<0.000
EURO-D score	2.32 (2.30, 2.34)	1.90 (1.87, 1.93)	2.63 (2.60, 2.66)	<0.000
EURO-D ≥ 4				<0.000
Yes [% (95% CI)]	25.3 (24.8, 25.7)	18.3 (17.8, 18.9)	30.5 (29.9, 31.1)	
No [% (95% CI)]	74.7 (74.3, 75.2)	81.7 (81.1, 82.2)	69.5 (68.9, 70.1)	
Hypertension				<0.000
Yes [% (95% CI)]	45.1 (44.7, 45.6)	44.9 (44.1, 45.6)	45.3 (44.7, 46.0)	
No [% (95% CI)]	54.9 (54.4, 55.3)	55.1 (54.4, 55.9)	54.7 (54.0, 55.3)	
MPA				<0.000
<1×/week	18.5 (18.1, 18.8)	17.6 (17.0, 18.2)	19.1 (18.6, 19.6)	
1/week	14.9 (14.6, 15.3)	15.0 (14.5, 15.5)	14.9 (14.4, 15.3)	
>1/week	66.6 (66.2, 67.1)	67.4 (66.7, 68.1)	66.0 (65.4, 66.6)	
VPA				<0.000
<1×/week	52.3 (51.8, 52.8)	48.5 (47.7, 49.2)	55.2 (54.6, 55.8)	
1/week	15.2 (14.8, 15.5)	15.0 (14.5, 15.5)	15.3 (14.9, 15.8)	
>1/week	32.5 (32.1, 33.0)	36.5 (35.8, 37.2)	29.5 (28.9, 30.1)	
CVE *				<0.001
Yes [% (95% CI)]	15.0 (14.7, 15.4)	18.4 (17.9, 19.0)	12.4 (12.0, 12.8)	
No [% (95% CI)]	85.0 (84.6, 85.3)	81.6 (81.0, 82.1)	87.6 (87.2, 88.0)	

Abbreviations: GS, grip strength; CVE, cardiovascular events; * CVE is history of cardiovascular events (myocardial infarction and/or stroke).

**Table 2 geriatrics-09-00036-t002:** Comparison of depressive symptoms according to the history of cardiovascular events (myocardial infarction and/or stroke).

	Depressive Symptoms (EURO-D 12 Score)						
	Total		Male		Female		*p*- Value
	Mean (SD)	Cohen’s *d*	Mean (SD)	Cohens *d*	Mean (SD)	Cohens *d*	
CVE history	3.01 (2.39)	2.14	2.53 (2.21)	1.93	3.55	2.23	<0.001
					(2.47)		
No CVE history	2.19 (2.09)		1.76 (1.85)		2.50 (2.19)		

Abbreviations: CVE, history of cardiovascular events (myocardial infarction and/or stroke); SD, standard deviation.

**Table 3 geriatrics-09-00036-t003:** Moderation analysis of grip strength for the relationship between cardiovascular events and depressive symptoms stratified by sex.

	Depressive Symptoms (EURO-D 12 Score)					
	Total Sample		Male		Female	
	B	95% CI	B	95% CI	B	95% CI
CVE (X)	1.05	0.89, 1.21	1.13	0.84, 1.42	1.03	0.74, 1.33
Grip Strength (W)	−0.04	−0.04, −0.04	−0.02	−0.03, −0.02	−0.05	−0.05, −0.04
CVE * Grip Strength	−0.01	−0.02, −0.01	−0.01	−0.02, −0.01	−0.01	−0.03, −0.01

Abbreviations: CI, confidence interval; CVE, cardiovascular events. * CVE is history of cardiovascular events (myocardial infarction and/or stroke).

## Data Availability

The data are freely accessible. The data can be accessed through the SHARE project website—https://share-eric.eu/ (accessed on 15 July 2023).

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
