# Peer review of "Moderating Effect of Muscular Strength in the Association between Cardiovascular Events and Depressive Symptoms in Middle-Aged and Older Adults—A Cross Sectional Study"

_geriatrics, 2024, doi:10.3390/geriatrics9020036_

Round 1
Reviewer 1 Report (Previous Reviewer 1)
Comments and Suggestions for Authors
Thank you for the new substantial revision of your manuscript and resubmitting it. In my opinion it has gained much in clarity and all of my comments have been addressed. The English language also has imprioved significantly. I appreciate the work you gave this paper and now recommend publication of the paper.
There is one typo I found in line 139: "present" should be "presented"
Author Response
Reviewer 1
Comment: Thank you for the new substantial revision of your manuscript and resubmitting it. In my opinion it has gained much in clarity and all of my comments have been addressed. The English language also has improved significantly. I appreciate the work you gave this paper and now recommend publication of the paper.
There is one typo I found in line 139: "present" should be "presented”.
Response: Thank you for your comments. We corrected the sentence as recommended.
Reviewer 2 Report (New Reviewer)
Comments and Suggestions for Authors
Title: Please consider that grip strength is just statistically a moderator, but grip strength is unable to have an effect on cardiovascular dimensions and depressive symptoms.
Please replace ‘events’ with a more specific and precise expression. Clarify, if the present study was cross-sectional or longitudinal. Further, indicate that you assessed a sample of elderly people.
Abstract: As mentioned, grip strength per se is unable to have any effect on any kind of physiological and psychological dimensions, but perhaps, grip strength is a proxy of a higher-order capacity. Please adjust and modify.
Introduction. “… share common pathogenic mechanisms…”; please specify and broaden this part, as it appears to be the backbone of your study.
“Several investigations have suggested..” is followed by just two references; please correct and amplify.
“…muscle strength is a plausible mechanism for these associations,…”; is muscle strengths really a directly impacting factor or a proxy of a healthier psychophysiological organism?
Overall, the Introduction section is very well written, though, some more precision is needed.
Methods. This part was nicely crafted.
Statistics: SPSS®, version 28.0 (IBM Corporation, Armonk NY; USA) for Windows®/Apple Mac®?
Table 1. Please add all statistical indices, including effect sizes. Keep in mind that reporting just p-values does not make sense (Wasserstein et al., 2019), as p-values basically get ‘significant’ with an increasing sample size, as this is the case of the present study (Cohen, 1988, 1992; Cohen, 1994; Zhu, 2012).
Results; this part is nicely crafted, though, it is mandatory to report the effect sizes (see also (Becker, 1988).
Discussion: This part was nicely crafted.
References
Becker, B.J., 1988. Synthesizing standardized mean-change measures. British Journal of Mathematical and Statistical Psychology 41(2), 257-278.
Cohen, J., 1988. Statistical power analysis for the behavioral sciences, 2nd ed. Lawrence Erlbaum Associates, Mahwah NJ.
Cohen, J., 1992. A power primer. Psychol Bull 112(1), 155-159.
Cohen, J., 1994. The earth is round (p < .05). American Psychologist 49(12), 997-1003.
Wasserstein, R.L., Schirm, A.L., Lazar, N.A., 2019. Moving to a World Beyond “p < 0.05”. The American Statistician 73(sup1), 1-19.
Zhu, W., 2012. Sadly, the earth is still round (P < 0.05). Journal of Sport and Health Science 1, 9–11.
Author Response
Reviewer 2
Comment: Title: Please consider that grip strength is just statistically a moderator, but grip strength is unable to have an effect on cardiovascular dimensions and depressive symptoms.
Response: Thank you for your comment. We changed to title to refer to muscular strength (a health-related component of physical fitness). As you mentioned, grip strength is just a measure of muscular strength.
Comment: Please replace ‘events’ with a more specific and precise expression. Clarify, if the present study was cross-sectional or longitudinal. Further, indicate that you assessed a sample of elderly people.
Response: Thank you for your comment on this topic. In the first submission of the manuscript, we used the term cardiovascular disease (CVD). However, one of the reviewers mentioned that “The authors seem to mix up CVD with cardiovascular events (CVE) which are not the same.” and suggested that we make that change.
Comment: Abstract: As mentioned, grip strength per se is unable to have any effect on any kind of physiological and psychological dimensions, but perhaps, grip strength is a proxy of a higher-order capacity. Please adjust and modify.
Response: Thank you for pointing this out. As mentioned in a previous answer, since we are analyzing muscular strength (a health-related component of physical fitness) and grip strength is just statistically a moderator, we also changed the abstract to comply with this notion.
Comment: Introduction. “… share common pathogenic mechanisms…”; please specify and broaden this part, as it appears to be the backbone of your study.
Response: Thank you for your suggestion. We have added information to the “introduction” section to clarify some of these mechanisms.
Comment: “Several investigations have suggested” is followed by just two references; please correct and amplify.
Response: Thank you for your comment. We have added more references to make the statement in question more robust.
Comment: “…muscle strength is a plausible mechanism for these associations…”; is muscle strengths really a directly impacting factor or a proxy of a healthier psychophysiological organism?
Response: Thank you for your insightful comment. Although a clear causal relationship cannot be demonstrated, there is data that suggests muscle strength may be a mediator or measure of general psychophysiological well-being in people with CVD. The link between physical and mental health may be explained by mechanisms that include the release of neurotrophic factors, such as brain-derived neurotrophic factor, and the control of inflammatory processes.
Comment: Methods. Statistics: SPSS®, version 28.0 (IBM Corporation, Armonk NY; USA) for Windows®/Apple Mac®?
Response: Thank you for your question. Apple Mac® was used. This information was added to the “Statistical Analysis” section.
Comment: Table 1. Please add all statistical indices, including effect sizes. Keep in mind that reporting just p-values does not make sense (Wasserstein et al., 2019), as p-values basically get ‘significant’ with an increasing sample size, as this is the case of the present study (Cohen, 1988, 1992; Cohen, 1994; Zhu, 2012).Results; this part is nicely crafted, though, it is mandatory to report the effect sizes (see also (Becker, 1988).
Response: Thank you for your suggestion. We added the requested information to the results section, namely table 1 and table 2.
This manuscript is a resubmission of an earlier submission. The following is a list of the peer review reports and author responses from that submission.
Round 1
Reviewer 1 Report
Comments and Suggestions for Authors
General:
The paper shows results of a study on moderating effects of grip strength on the well-known relation between depressive symptoms and Cardiovascular disease (CVD). The methods include only few variables and are very basic with major faults concerning the category of CVD: Inclusion criteria was any self-reported history of CVE (stroke or myocardial infarction) without any data on the number of events, time since the event and medical comorbidities such as hypertension, coronary artery disease… The authors seem to mix up CVD with cardiovascular events (CVE) which are not the same. also history of depression was not obtained although this would be another important covariate.
Results are displayed appropriately but there is quite some redundancy between text and tables (table 2 and 3, which are already well known correlations without additional value of this paper.
The discussion has several redundancies and, in some parts, explains trivial factors (such as what a negative association means) but neglects the broad body of evidence on positive effects of physical activity (endurance and strength) on depression and depressive symptoms in general as well as in populations with cardiovascular disease. The observed moderating effect of GS – although statistically significant – seems to be extremely small in comparison with the direct effect of CVD on depressive symptoms. This is not discussed at all. The effect size does not support the conclusion according to my understanding of the applied statistical method. Limitations are discussed superficially and in no structured order.
Introduction:
(1) Main symptoms of depression according to ICD-11 are depressed mood and diminished interest. Please refer to common diagnostic system for the definition of depression (ICD-11 or DSM-5) and use the corresponding terms.
(2) Lifetime risk depression is higher according to other studies not included in the cited meta-analysis, i.e:
o Hasin DS, Goodwin RD, Stinson FS, Grant BF. Epidemiology of major depressive disorder: results from the National Epidemiologic Survey on Alcoholism and Related Conditions. Arch Gen Psychiatry. 2005 Oct;62(10):1097-106.
o Bromet E, Andrade LH, Hwang I, Sampson NA, Alonso J, de Girolamo G, et al. Cross-national epidemiology of DSM-IV major depressive episode. BMC Med. 2011 Jul 26;9:90.
Therefore, please state that lifetime risk is at least 10% or define an appropriate range, since there is also significant variance between different countries.
(3) Line 45: Since CVD is not an only cause of depression please change the wording. I.e.: “…is an important risk factor”. The same goes for depression “causing” CVD. Depression is one of many risk factors for CVD.
Methods:
According to your methodology you measured 4 values for grip strength (GS) in each participant but you don’t define how you proceeded with those measurements. How did you define GS as variable per person? Use of the best measurement (highest value)? Mean of all 4 measures? Did you record dominant hand? The paper you referenced proposes 3 measures per hand and using the best result as variable for GS. The authors also suggest monitoring hand dominance.
Statistics:
Please define how you report data more accurately. I.e. “frequency” can only be reported for categorical data, not continuous measures such as age or depressive symptoms.
Tables and Figures:
Figure 1 does not give any additional information and should either be excluded or be shown instead of table 4 with the values of moderation analysis. I believe it is also more common to show separate paths between all 3 variables (Main: X – Y, moderators: X – W and W – Y) all with their respective b-value.
Comments on the Quality of English LanguageThere are several major language-related issues that make the paper difficult to read and comprehend.
Author Response
Comment: The paper shows results of a study on moderating effects of grip strength on the well-known relation between depressive symptoms and Cardiovascular disease (CVD). The methods include only few variables and are very basic with major faults concerning the category of CVD: Inclusion criteria was any self-reported history of CVE (stroke or myocardial infarction) without any data on the number of events, time since the event and medical comorbidities such as hypertension, coronary artery disease… The authors seem to mix up CVD with cardiovascular events (CVE) which are not the same. also history of depression was not obtained although this would be another important covariate.
Response: Thank you for your comments, which have helped improve the quality of our manuscript. We understand your concerns about the definition of CVD used in our study. Following your suggestion, we changed the term CVD to cardiovascular events where appropriate. Additionally, we added the number of events to the results section and hypertension to the covariables. We have also added the lack of controlling variables and the self-reported nature of the outcome to the limitations section.
Comment: Results are displayed appropriately, but there is quite some redundancy between text and tables (table 2 and 3, which are already well known correlations without additional value of this paper.
Response: Thank you for your comment. We improved the results text to reduce the redundancy between text and tables. Also, we opted to maintain the results of table 2 in text only, thus deleting table 2.
Comment: The discussion has several redundancies and, in some parts, explains trivial factors (such as what a negative association means) but neglects the broad body of evidence on positive effects of physical activity (endurance and strength) on depression and depressive symptoms in general as well as in populations with cardiovascular disease.
Response: Thank you for your comment. We have improved the discussion accordingly.
Comment: The observed moderating effect of GS – although statistically significant – seems to be extremely small in comparison with the direct effect of CVD on depressive symptoms. This is not discussed at all. The effect size does not support the conclusion according to my understanding of the applied statistical method. Limitations are discussed superficially and in no structured order.
Response: Thank you for pointing this out. We have altered the text to adequate the discussion and conclusion to our findings, namely the small effect size of the moderation effect. Also, we have improved the limitations section to consider the downsides of our study strongly.
Comment: (1) Main symptoms of depression according to ICD-11 are depressed mood and diminished interest. Please refer to common diagnostic system for the definition of depression (ICD-11 or DSM-5) and use the corresponding terms.
Response: Thank you for your suggestion. The DSM-5 criteria for depression diagnosis were added to the introduction.
Comment: (2) Lifetime risk depression is higher according to other studies not included in the cited meta-analysis, i.e:
o Hasin DS, Goodwin RD, Stinson FS, Grant BF. Epidemiology of major depressive disorder: results from the National Epidemiologic Survey on Alcoholism and Related Conditions. Arch Gen Psychiatry. 2005 Oct;62(10):1097-106.

o Bromet E, Andrade LH, Hwang I, Sampson NA, Alonso J, de Girolamo G, et al. Cross-national epidemiology of DSM-IV major depressive episode. BMC Med. 2011 Jul 26;9:90.
Therefore, please state that lifetime risk is at least 10% or define an appropriate range, since there is also significant variance between different countries.
Response: Thank you for your comment. Data about the lifetime risk of depression was added to the introduction.
Comment: (3) Line 45: Since CVD is not an only cause of depression please change the wording. I.e.: “…is an important risk factor”. The same goes for depression “causing” CVD. Depression is one of many risk factors for CVD.
Response: Thank you for your comment. The wording was changed accordingly.
Comment: According to your methodology you measured 4 values for grip strength (GS) in each participant but you don’t define how you proceeded with those measurements. How did you define GS as variable per person? Use of the best measurement (highest value)? Mean of all 4 measures? Did you record dominant hand? The paper you referenced proposes 3 measures per hand and using the best result as variable for GS. The authors also suggest monitoring hand dominance.
Response: Thank you for your comment on this topic. The clarification about how the GS variable was generated was added to the methodology. Although the referenced paper proposes 3 measures per hand, we used the same methodology as the SHARE protocol.
Comment: Please define how you report data more accurately. I.e. “frequency” can only be reported for categorical data, not continuous measures such as age or depressive symptoms.
Response: Thank you for your comment. More information was added to the statistical analysis to clarify the data procedure differences with continuous and categorical variables.
Comment: Figure 1 does not give any additional information and should either be excluded or be shown instead of table 4 with the values of moderation analysis. I believe it is also more common to show separate paths between all 3 variables (Main: X – Y, moderators: X – W and W – Y) all with their respective b-value.
Response: Thank you for your suggestion. We have deleted figure 1 as the moderation statistical model is widely known.
Reviewer 2 Report
Comments and Suggestions for Authors
This manuscript titled “ Moderating effect of grip strength in the association between cardiovascular disease and depressive symptomatology” aimed to explore the relationship between grip strength, cardiovascular diseases and depressive symptoms among middle-aged and older adults. The manuscript needs some more clarification.
Materials and Methods: please provide the aged in the inclusion criteria. Please provide the definition of CVD and please add in the characteristic data. Did you report the max. GS or average GS? you did 2 times in both hand; please provide the details.
Did you consider regarding the memory or any confounding factors that might affect GS or depression?
Please provide the physiological or mechanisms of GS and depression in CVD in the discussion part.
CVD has been defined in several diseases and that might affect GS or depression and the duration or severity of CVD also relate to GS or depression. Please discussion or provide the details in the results and discussion part.
Comments on the Quality of English Language
English can be improved.
Author Response
Comment: This manuscript titled “Moderating effect of grip strength in the association between cardiovascular disease and depressive symptomatology” aimed to explore the relationship between grip strength, cardiovascular diseases and depressive symptoms among middle-aged and older adults. The manuscript needs some more clarification.
Response: Thank you for your comments, which have helped improve the quality of our manuscript. We have modified the manuscript according to your comments.
Comment: please provide the aged in the inclusion criteria. Please provide the definition of CVD and please add in the characteristic data. Did you report the max. GS or average GS? you did 2 times in both hand; please provide the details.
Response: Thank you for your comment. We have added the age to the inclusion criteria and more information about the GS data collection and measurement. About the definition of CVD, as suggested by reviewer 1 we altered it to cardiovascular events (myocardial infarction and stroke) as it gives a better picture of our measure of CVD.
Comment: Did you consider regarding the memory or any confounding factors that might affect GS or depression?
Response: Thank you for your insightful comment. We have added this to the limitations section.
Comment: Please provide the physiological or mechanisms of GS and depression in CVD in the discussion part. CVD has been defined in several diseases and that might affect GS or depression and the duration or severity of CVD also relate to GS or depression. Please discussion or provide the details in the results and discussion part.
Response: Thank you for your suggestion. The physiological mechanisms that could explain the complex connections between depression, CVD and GS have been added to the discussion part.
Reviewer 3 Report
Comments and Suggestions for Authors
Introduction
1) Beginning on line 44: The association between CVD and depression has been investigated quite a bit. Based upon the existing literature, you might want to consider describing the association in stronger terms and then provide more key references.
2) Does the article below strengthen your argument?
Title: A combination of depression and decreased physical function further worsens the prognosis of patients with chronic cardiovascular disease
Authors: Kenya Osada, Minako Yamaoka-Tojo*, Shinichi Obara, Hidenori Kariya, Yohei Kato, Akinori Yuyama, Kentaro Kamiya, Atsuhiko Matsunaga and Junya Ako
3) Muscle dysfunction? What is meant by this term? Weakness, sarcopenia etc.? Is there a clear clinical definition?
4) Related to comment above, it seems you are assessing muscle strength not dysfunction. I would replace “dysfunction” with weakness or something that more clearly represents the GS outcome.
Materials and Methods
5) Line 78: “The exposure measure…” Unclear sentence. Maybe change to, “The exposure measure was defined as a previous Dx of CVD…
6) Lines 84 – 89 could be written more clearly.
7) Regarding your conceptual model, I am wondering about the bidirectional association between CVD and depression. It seems relatively established that this bidirectional association exists. Do your statistical methods account for that association? Should both models be tested?
8) Table 2: Correlation between GS and Depressive Symptoms
These are not the strongest correlations, perhaps significance is being driven by sample size? Interpretation may require some caution.
Discussion
9) Comment about overall findings.
- I think this paper somewhat simplifies the association between muscle strength and CVD/Depression. Within the Discussion section, I feel it necessary to discuss the possibility that other more complex factors that include muscle strength will exert an influence on the association in question. One good example is frailty. We know that grip strength is an important component of the frailty phenotype. It may be that frailty then influences CVD/Depression but your model does not control for frailty status.
- “Even with an elderly person who continues to be at risk of being diagnosed with sarcopenia, interventions that increase their strength will immediately have a protective effect against depression, attenuating its symptoms”.
This statement is too bold given the complexity of the factors that are associated with depression. Yes, I would agree that improved physical fitness overall, because of increased physical activity participation is clearly good to reduce both CVD and depression risk, however, your assertion of the immediate protective impact is not really supported by this cross-sectional investigation.
- Line 172: Unclear what this sentence is stating.
- “Overall, this study's findings can provide valuable insights for healthcare practitioners, policymakers, and researchers to develop targeted strategies that address the complex interplay between mental and cardiovascular health and promote better overall well-being.”
What insights? I think this needs elaboration.
- I think it is important to mention that your model does not adjust for any covariates that likely have an influence on these associations. I provided the frailty idea but there are others that should be acknowledged. Is GS acting as a proxy for something other than simple overall muscle strength?
Comments on the Quality of English Language
Please review for clarity. There are a few minor issues.
Author Response
Comment: 1) Beginning on line 44: The association between CVD and depression has been investigated quite a bit. Based upon the existing literature, you might want to consider describing the association in stronger terms and then provide more key references.
Does the article below strengthen your argument?
Title: A combination of depression and decreased physical function further worsens the prognosis of patients with chronic cardiovascular disease
Authors: Kenya Osada, Minako Yamaoka-Tojo*, Shinichi Obara, Hidenori Kariya, Yohei Kato, Akinori Yuyama, Kentaro Kamiya, Atsuhiko Matsunaga and Junya Ako
Response: Thank you for your suggestion. We have added this to the introduction.
Comment: 2) Muscle dysfunction? What is meant by this term? Weakness, sarcopenia etc.? Is there a clear clinical definition? Related to comment above, it seems you are assessing muscle strength not dysfunction. I would replace “dysfunction” with weakness or something that more clearly represents the GS outcome.
Response: Thank you for your insight on this. The word dysfunction was replaced with weakness.
Comment: 3) Line 78: “The exposure measure…” Unclear sentence. Maybe change to, “The exposure measure was defined as a previous Dx of CVD…
Response: Thank you for your comment. This sentence was already changed according to a suggestion of reviewer 1.
Comment: 4) Lines 84 – 89 could be written more clearly.
Response: Thank you for your comment. The sentences were reviewed and improved to be clearer.
Comment: 5) Regarding your conceptual model, I am wondering about the bidirectional association between CVD and depression. It seems relatively established that this bidirectional association exists. Do your statistical methods account for that association? Should both models be tested?
Response: Thank you for your comment. Our statistical model does not establish direction and thus accounts for the bidirectional associations that exits between all variables in the model. This is a limitation of the study that is reported in the limitations section.
Comment: 8) Table 2: Correlation between GS and Depressive Symptoms
These are not the strongest correlations, perhaps significance is being driven by sample size? Interpretation may require some caution.
Response: Thank you for signalling this out. Following your and reviewer 1 comment, we have altered the text substantially to consider the small effect sizes of the findings. Notwithstanding, the associations found show a clear trend in the nefarious and beneficial effects of CVD and GS on depression.
Comment: 9) Comment about overall findings.
- I think this paper somewhat simplifies the association between muscle strength and CVD/Depression. Within the Discussion section, I feel it necessary to discuss the possibility that other more complex factors that include muscle strength will exert an influence on the association in question. One good example is frailty. We know that grip strength is an important component of the frailty phenotype. It may be that frailty then influences CVD/Depression but your model does not control for frailty status.
Response: Thank you for your comment. We have improved the discussion in its different paragraphs on this topic (complex association between GS, CVD and depression). Regarding frailty, we do not have information to control for this variable. However, GS is one of the criteria for frailty status, and its inclusion as one of the main variables serves as a proxy of the association between frailty and CVD and depression.
Comment: - “Even with an elderly person who continues to be at risk of being diagnosed with sarcopenia, interventions that increase their strength will immediately have a protective effect against depression, attenuating its symptoms”.
This statement is too bold given the complexity of the factors that are associated with depression. Yes, I would agree that improved physical fitness overall, because of increased physical activity participation is clearly good to reduce both CVD and depression risk, however, your assertion of the immediate protective impact is not really supported by this cross-sectional investigation.
Response: Thank you for your comment. We agree with you. The sentence was toned down to consider the multifactorial nature of depression.
Comment: - Line 172: Unclear what this sentence is stating.
Response: Thank you for your comment. The sentence was reviewed and improved to be clearer.
Comment: - “Overall, this study’s findings can provide valuable insights for healthcare practitioners, policymakers, and researchers to develop targeted strategies that address the complex interplay between mental and cardiovascular health and promote better overall well-being.”
What insights? I think this needs elaboration.
Response: Thank you for your comment. We agree that the sentence was not clear and eliminated it.
Comment: - I think it is important to mention that your model does not adjust for any covariates that likely have an influence on these associations. I provided the frailty idea but there are others that should be acknowledged. Is GS acting as a proxy for something other than simple overall muscle strength?
Response: Thank you for your comment. We have added a more in-depth list of variables that may confound this association to the limitations section, acknowledging its importance. Yes, GS may also be a proxy of frailty, as it is one of its criteria. Thus, this also means that people with less frailty (at least in one of its criteria) are slightly protected in the association between CVD and depression. However, because this is an exaggerated extrapolation of the findings, we did not explore this in the manuscript.
Reviewer 4 Report
Comments and Suggestions for Authors
Overall, the methodology and results are difficult to understand. Mainly because the objective of the article is to identify the role of handgrip strength as a moderator in the relationship between cardiovascular diseases and depressive symptoms, however the way in which its analysis is conducted and exposed is always divided by the gender, which does not appear in its objectives. Thus, the tables should display gender/sex data as a possible moderating variable and not as an independent variable, because none of its outcomes were organized using it. Another point to be considered is that from the moment the authors decide to carry out path and moderation analyses, it is necessary to assume that there are other variables participating in this relationship. Thus, variables such as gender/sex, income, age, access to medical care, nutritional status, level of physical activity are all possible moderators and need to be taken into account in the model. I suggest making a causal diagram with the hypotheses of the authors. The database used has rich information that may also be playing an important role in the analyzed outcomes.
Introduction:
The introduction well describes the association between cardiovascular diseases and depressive symptoms, however, little is explored about the relationship between muscle strength/dynapenia and the evaluated outcomes.
Methods:
Figure 1: I suggest a revision of figure 1. In the formulation of a causal diagram of the variables, the handgrip strength variable is associated by itself with the symptomatology of depression and also with cardiovascular diseases, so it is a moderator. Thus, it is necessary to make this relationship clearer in the figure, in which two arrows should depart from handgrip strength, one for depressive symptoms and the other for cardiovascular diseases.
Results:
Table 1: I suggest inserting the significance values
Comments on the Quality of English Languagenone
Author Response
Comment: Overall, the methodology and results are difficult to understand. Mainly because the objective of the article is to identify the role of handgrip strength as a moderator in the relationship between cardiovascular diseases and depressive symptoms, however the way in which its analysis is conducted and exposed is always divided by the gender, which does not appear in its objectives. Thus, the tables should display gender/sex data as a possible moderating variable and not as an independent variable, because none of its outcomes were organized using it.
Response: Thank you for your comments, which have helped improve the quality of our manuscript. We have improved the methods and results section to be clearer, detailed and concise. Also, we stratified the analysis by sex because it is a well-known determinant of CVD and depression. We added this to the aims of our study.
Comment: Another point to be considered is that from the moment the authors decide to carry out path and moderation analyses, it is necessary to assume that there are other variables participating in this relationship. Thus, variables such as gender/sex, income, age, access to medical care, nutritional status, level of physical activity are all possible moderators and need to be taken into account in the model. I suggest making a causal diagram with the hypotheses of the authors. The database used has rich information that may also be playing an important role in the analyzed outcomes.
Response: Thank you for your comment. In our study, due to its cross-sectional nature, it is impossible to infer causality and direction. Thus, the statistical model is bidirectional and considers the existing associations between all variables. Also, following other reviewers’ suggestions, we included hypertension in the model as the major covariable of cardiovascular events and have modified the discussion and limitations sections to consider the multifactorial determinants of CVD and depression and the lack of control in the study (one of the major limitations).
Comment: The introduction well describes the association between cardiovascular diseases and depressive symptoms, however, little is explored about the relationship between muscle strength/dynapenia and the evaluated outcomes.
Response: Thank you for your comment. We have improved the introduction according to your suggestion.
Comment: Figure 1: I suggest a revision of figure 1. In the formulation of a causal diagram of the variables, the handgrip strength variable is associated by itself with the symptomatology of depression and also with cardiovascular diseases, so it is a moderator. Thus, it is necessary to make this relationship clearer in the figure, in which two arrows should depart from handgrip strength, one for depressive symptoms and the other for cardiovascular diseases.
Response: Thank you for your suggestion. The diagram in figure 1 refers to the statistical model of moderation used in this study. As suggested by reviewer 1, we have deleted figure 1 as the moderation statistical model is widely known.
Comment: Table 1: I suggest inserting the significance values
Response: Thank you for your suggestion. We have performed t tests for continuous variables and chi-square for categorical variables and reported the p-value in Table 1.
Round 2
Reviewer 1 Report
Comments and Suggestions for Authors
The manuscript must be edited by either an English native speaker or a preofessional editing service.
Reviewer 2 Report
Comments and Suggestions for Authors
The revised manuscript can be accepted.
Comments on the Quality of English LanguageMinor editing of English language is required, please check the format of the references.
Reviewer 3 Report
Comments and Suggestions for Authors
No additional changes.
Reviewer 4 Report
Comments and Suggestions for Authors
1- INTRODUCTION section:
General commentary: I believe that the authors should use the introduction to further explore issues related to gender, dynapenia/CVC and depression, so that this relationship does not appear only in the objective of the text.
2- METHODS and Results section
General commentary: Despite the enthusiasm in carrying out the requested review and the general improvement of the article, the problems with the methodology and results still remain. It is very difficult to understand the plausibility and reproducibility of the results present in the article, mainly because the authors addressed a major problem (the possible moderating effect of handgrip strength on the association between cardiovascular diseases and depressive symptoms), which has many moderating and confounding variables and assumed that the result found was really just the relationship between these three variables. The article continues to present no data, no effort to control or understand from a statistic perspective the role that other variables such as age, income, nutritional status and level of physical activity can play in the relationship studied by them. For example, one of the main syndromes studied during aging is frailty, which has a high prevalence in older adults and involves depressive symptoms and hand grip strength. This association found may also be moderated by frailty, especially considering that these individuals were not identified or excluded. All of this means that the article, despite having an important question and a large sample, has very profound methodological limitations. Therefore, once again I suggest that authors think about their question and the factors that may be moderating this association. Observed what can be controlled, whether in the methodology (excluding some conditions), through statistical analysis or even been cited in the discussion or in the limitation section of the article.
Comments on the Quality of English Languagenone